# Does Spirituality Influence Happiness and Academic Performance?

**Rajasekhar David** [1], **Sharda Singh** [2,*] , **Neuza Ribeiro** [3] and **Daniel Roque Gomes** [4]

1 Indian Institute of Management, Ranchi 834002, India; rajasekhar.david@iimranchi.ac.in
2 Xavier Institute of Social Service, Ranchi 834002, India
3 CARME-Centre of Applied Research in Management and Economics ESTG, Polytechnic Institute of Leiria, 2411-901 Leiria, Portugal; neuza.ribeiro@ipleiria.pt
4 School of Education, Polytechnic Institute of Coimbra, ICNOVA–Instituto de Comunicação da NOVA, 1069-061 Lisboa, Portugal; drmgomes@ipc.pt
* Correspondence: shardasingh@xiss.ac.in

**Abstract:** One of the key issues of the learning experience is students' performance during the course, as this is pointed to as one of the main indicators for boosting competences' development and skills' improvement. This study explores the roles of spirituality, forgiveness, and gratitude on students' academic performance, proposing a model of analysis revealing a first-order moderation effect of spirituality in the mediation effect of happiness, on the relation between gratitude and forgiveness with students' academic performance. Two hundred twenty management students from various Indian universities voluntarily participated in the study. To avoid common method-bias issues, data concerning the study variables were obtained in two distinct moments. To test for the moderated-mediation model of analysis, we have followed the PROCESS analytical procedure. Results showed that forgiveness and gratitude were positively and significantly related to happiness and academic performance. It was also possible to see that spirituality moderates the relationship between forgiveness for self and student happiness. Finally, the moderated-mediating impact of spirituality and happiness on the relationship between gratitude and academic performance was also supported. The present study has taken the lead from positive psychology to assess the students' character strengths related to their well-being and success. It proposes an innovative model of analysis, supported by theoretical reasoning, pointing to the existence of a moderated-mediation relation predicting students' academic performance.

**Keywords:** forgiveness; gratitude; spirituality; happiness; academic performance

## 1. Introduction

With the increasing number of multinational corporations in India, organizations' demand for managerial professionals has grown substantially. Management has become a popular choice for young graduates in India (Jagadeesh 2000). Taking the lead from positive psychology, students' character seems to be a relevant issue to attend to for improving students' academic performance. In 2000, Seligman and Csikszentmihalyi initiated an approach regarding students' academic performance, based on positive aspects of human life, realizing the importance of human strengths to improve students' academic performance such as spirituality, gratitude, happiness, and forgiveness, which buffer against negative consequences such as depression and suicidal tendencies (Hirsch et al. 2007).

In a survey of 47 countries, students rated happiness as a top priority, even ahead of love and health (Diener 2000). Perhaps this is due to students considering happiness a top priority for producing rapid life benefits, including effective functioning and good health (Diener 2012). Existing research seems to show a trend of research focused on understanding the predictors of the positive aspects of students' happiness (Seligman et al. 2009). The positive emotions produced by happiness seem essential in broadening an individual's thought actions, enabling an individual to think creatively and to become

more resilient toward problems (Fredrickson 2004). It has been found that happiness is an essential emotional attribute in academic performance (Pekrun et al. 2002), suggesting that happy students will perform better in their academic examinations and will try harder for a better career once they have graduated. Consequently, enhancing happiness among college students will probably promote effective learning among students.

Peterson and Seligman (2004) identified six common universal virtues across a broad sample of cultures, religions, and moral philosophers: wisdom, courage, humanity, justice, temperance, and transcendence. They have identified numerous character strengths that exemplify these virtues and have passed a battery of validity criteria. Character strengths are essential, as a well-lived and happy life is strongly associated with human virtues (Peterson and Seligman 2004). Forgiveness, gratitude, and spirituality are listed in the 24 character strengths and six core virtues. This study is the first of its kind, including four crucial character strengths for improving students' performance. Character education may offer a solution to young students' depression, as it enhances learning and creativity and promotes civic citizenship (Waters 2011). Therefore, the present study explored the effects of forgiveness and gratitude on happiness and academic performance in the presence of spirituality.

## 2. Theoretical Background

The lion's share of the research into forgiveness until now has focused only on the negative consequences of a failure to forgive. In recent years, research on forgiveness has grown (Mccullough 2000). Researchers commonly agree that forgiveness is giving up one's right to retaliation or letting go of a negative affect toward the transgressor. Forgiveness has been further operationalized in terms of the context where it is performed: the forgiveness of self, forgiveness of situations, and forgiveness of others (Thompson et al. 2005). Forgiveness for self is defined as a psychological process where an individual tries to replace their negative cognitions, emotions, and behavior (e.g., guilt, shame, sadness) with positive cognitions, emotions, and behavior. Forgiveness of others is defined as a process of constructive change in one's cognitions, emotions, and behavior toward a transgressor. Forgiveness of the situation is defined as a psychological process of changing negative emotions to positive emotions regarding events that an individual views as beyond their control, such as illness and natural disaster.

Forgiveness is also considered a positive psychological response to interpersonal harm. Individuals who perform a more significant act of forgiveness are likely to experience a higher level of happiness. Several studies have revealed that individuals high on forgiveness have better health and tend to be happier than individuals low on forgiveness (Hannon et al. 2012). Forgiveness can reduce the adverse effects of hatred, which leads to excessive stress, and forgiveness leads to higher life satisfaction. Forgiveness is also closely associated with spirituality. Previous studies have shown that spirituality has beneficial effects on physical and mental health. Forgiveness is frequently considered a specific characteristic of spirituality in most of the religions (Davis et al. 2013; Stoycheva 2018). There has been a growing interest in spirituality in the academic literature in the past few decades. Despite a great deal of research on spirituality, researchers have reached little consensus on what spirituality means. The terms "spirituality" and "religiousness" are used inconsistently and interchangeably by researchers. Spirituality should be better understood as a latent variable that has multiple dimension that make it one (Nolan 2011). Spirituality can be defined as a dynamic and intrinsic aspect of humanity, through which a person seeks ultimate meaning, purpose, and transcendence and experiences a relationship with self, family, other, community, society, nature, and the significant or sacred (Puchalski et al. 2009), while religion represents both an individual and an institutional construct. Religion is a fixed system of ideas or ideological commitments that do not address the subjective components; on the other hand, spirituality refers to a personal, subjective side of a religious experience.

The positive association between forgiveness and well-being may function through direct and indirect mechanisms. The primary function may operate through the cognitive process of rumination, producing negative emotions such as hatred and anger (Levens et al. 2009). An indirect effect may function through associations with constructs such as well-being, social support, and interpersonal functioning (Webb et al. 2011). People inclined to spirituality tend to forgive their transgressors and tend to be more agreeable, emotionally stable, and happier than people who are not inclined to spirituality (Davis et al. 2013). The roots of happiness studies can be traced to the work of great philosophers of ancient Greece, such as Aristotle, but empirical investigation was lacking until the 1960s. Wilson (1967) proposed that happiness is affected by access to basic needs and satisfaction with fulfilling those needs. However, Wilson's (1967) model has not incorporated the subjective evaluation of individuals in understanding happiness. Subjective well-being (SWB) refers to an individual's judgment about how satisfying and fulfilling their life is. SWB is a higher-order construct consisting of effect and subjective evaluation of one's life. SWB is also known as happiness in colloquial terms (Diener 2000). Happiness results in experiencing a predominance of positive affect most of the time, a rare experience of negative affect, and higher life satisfaction (Lyubomirsky et al. 2005). Life satisfaction is conceptualized as one's attitude, belief, and judgment toward their life. Happy people differ from their less-happy peers in perceiving and reacting adequately to various life events (Lyubomirsky 2001). A meta-analysis conducted by Lyubomirsky et al. (2005) reported that positive emotions induce success. Many benefits are associated with happiness, such as longevity, lasting relationships, earning more money, etc.

Gratitude, on the other hand, is derived from the Latin word 'gratia', meaning gratefulness or graciousness. All derivatives of this Latin word have something to do with kindness or the beauty of receiving and giving. Gratitude has been conceptualized as an emotion, a moral virtue, an attitude, a coping response, or a personality trait. The object of gratitude can be humans or non-humans such as nature, animals, and God (Solomon 1976). Gratitude has been defined as "the willingness to recognize the unearned increments of value in one's experience" (Bertocci and Millard 1963), and "an estimate of gain coupled with the judgment that someone else is responsible for that gain" (Solomon 1976). The benefit or gain can be spiritual or emotional. As an emotion, gratitude is an attribution-dependent state that has an outcome of two-step cognitive processes: (a) noticing that one has acquired a positive outcome, and (b) noticing an external cause for this positive outcome.

Gratitude is often compared to moral emotions because of its following three functions: (1) It elicits a response in a beneficiary to react to the moral actions performed by the benefactor. (2) It motivates the beneficiary to react prosocially toward others. (3) The prosocial behavior exhibited by the beneficiary reinforces the benefactor to exhibit moral behavior in future. Gratitude is strongly related to numerous positive outcomes such as superior life satisfaction and relationship satisfaction. Gratitude interventions enhance positive emotions and interpersonal trust (Drążkowski et al. 2017). Fostering gratitude reduces materialism in children and adolescents (Chaplin et al. 2019).

The cognitive and psychosocial frameworks based on Fredrickson's broaden-and-build theory suggest that gratitude is associated with increased happiness (Alkozei et al. 2018). Fredrickson's (2001) broaden-and-build theory of positive emotions suggests that gratitude, like any other positive emotion, may also help individuals build other durable resources for well-being. Specifically, it may foster creativity, intrinsic motivation, and purposefulness (Bono et al. 2008) as well as spark an upward spiral of positive emotions and outcomes. Many researchers found a significant relationship between gratitude and happiness (Hill and Allemand 2011). The ability to notice and be grateful for the events of one's life has been considered an essential element of happiness.

It is also be worth noting that gratitude and spirituality are also related to one another. Many religious traditions place a high value on gratitude. The broaden-and-build theory explains why gratitude would be positively linked to spirituality. When grateful, the mindset of an individual is broadened to include the role others play in supporting their

well-being. Happiness can be proposed as the ultimate human gratification, if spirituality and gratitude can be cataloged as virtues and human strengths (Peterson and Seligman 2004). Happiness among grateful people makes them more involved and interested in people, events, and work around them, consequently improving the individual's performance in various domains of life. In addition to gratitude, spirituality believes that a loving and guiding divine force provides the meaning and purpose of an individual's life. Thus, an individual high in spirituality may use gratitude more frequently (Rosmarin et al. 2010), enhancing their happiness and improving their performance in various life domains. Therefore, it is hypothesized that spirituality moderates the relationship between forgiveness and happiness as well as between gratitude and happiness (see Figure 1). Therefore, and according to the assumptions previously presented, we propose the following hypotheses:

**H1a.** *Spirituality moderates the relationship between forgiveness for self and happiness.*

**H1b.** *Spirituality moderates the relationship between forgiveness for others and happiness.*

**H1c.** *Spirituality moderates the relationship between forgiveness for situation and happiness.*

**H2.** *Spirituality moderates the relationship between gratitude and happiness.*

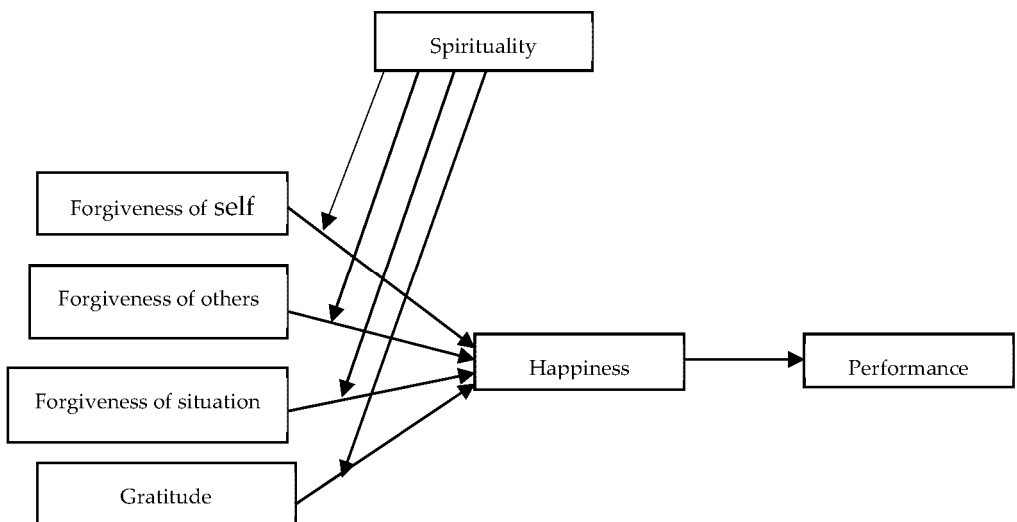

**Figure 1.** Conceptual model.

Past research connected to happiness (Seligman et al. 2009) suggests an inclination toward students' learning. The positive emotions produced by happiness are desirable to broaden an individual's thought actions, enabling an individual to think creatively and become more resilient toward problems (Fredrickson 2004). It has been found that happiness is an essential factor in academic performance (Pekrun et al. 2002). This suggests that happy students will perform better in their academic examinations and try harder for a better career once they have graduated. Consequently, developing happiness among college students promotes effective learning (O'Donnell and Gray 1993).

When individuals forgive a transgressor, their revenge and aggression-related motivations subside, and their push toward goodwill and compassion increases. Forgiveness comes from a desire to restore someone's goodwill and ignorance toward negative emotional states. Forgiveness improves physical health as well as psychological health. Several studies in the religious literature have noticed the positive relationship between forgiveness and spirituality (Bureau 2013). Langman and Chung (2013) found a positive connection between spirituality, forgiveness, and psychological well-being in a sample of drug and alcohol addicts. Spirituality is more likely to improve inner strengths such as forgiveness and encourage individuals to find peace and happiness in stressful circumstances (Arévalo et al. 2008). It also provides an optimistic perspective and positive purpose in life, which helps to enhance the effectiveness in a different arena of life.

Emmons and McCullough (2003) defined gratitude as a cognitive-affective state where an individual perceives that their success was not deserved and earned but due to the good intentions of others. Gratitude enhances social relationships, kind attitudes toward others, goodwill, and overall personal well-being. Therefore, it serves as a moral barometer and reinforces happiness. The existing literature does not provide an adequate relationship between the constructs' gratitude, spirituality, and happiness. Emmons and McCullough (2003) stated that gratitude serves as a gap between spirituality and religion. Spirituality can be conceptualized as a moderator because it provides a meaningful system that influences an individual's life, highlighting the importance of gratitude. Gratitude leads to happiness (Bono et al. 2008), and, subsequently, happiness leads to superior academic performance (Pekrun et al. 2002). Therefore, and according to the assumptions previously presented, we propose the following hypotheses (see Figure 1):

**H3a.** *Spirituality moderates the indirect relationship between forgiveness for self and performance through happiness, such that this indirect relationship is stronger at a high level of spirituality.*

**H3b.** *Spirituality moderates the indirect relationship between forgiveness for others and performance through happiness, such that this indirect relationship is more potent at a high level of spirituality.*

**H3c.** *Spirituality moderates the indirect relationship between forgiveness for situation and performance through happiness, such that this indirect relationship is more potent at a high level of spirituality.*

**H4.** *Spirituality moderates the indirect relationship between gratitude and academic performance through happiness, such that this indirect relationship is more potent at a high level of spirituality.*

## 3. Materials and Methods

### 3.1. Procedure Overview

The sampling design for our study is non-probabilistic. All the participants were part of an art of living workshop, organized by the one of the business schools as part of an extra-curricular activity. The workshop included sessions on spirituality and mindfulness. To have a data collection procedure immune to common-method bias, data collection occurred in two distinct moments (before and after the workshop). All the respondents were informed that there would be two stages of data collection. Accordingly, data were collected at two-time points with a time lag of 3 months. Participants were also informed that the data collected would be kept confidential and used only for research purposes. A self-reported survey instrument containing measures on spirituality, gratitude, forgiveness, happiness, and demographic variables was included and distributed among 300 second-year MBA students at time 1. The academic performance of the second semester was retrieved from the examination controller of the university. The measurement instrument consisted of 44 items. At time 1, left after initial data cleaning, the final usable response was 220. There was an almost equal proportion of male (50.9%) and female (49.1%) respondents in the sample. The majority of respondents (55%) reported their family income between INR 500,000–INR 1,000,000. The mean age of the respondents was 22.3 years.

The responses were coded with the last four digits of the participants' contact numbers. After three months, the 220 participants were approached to complete the same questionnaire again; only third-semester grades were retrieved at time 2. In total, 186 responses were returned in the stipulated time, out of which 174 responses were usable. The demographic profile of time 2 was very similar to time 1 data.

### 3.2. Measures

*Forgiveness* was measured through the 18-item Heartland Forgiveness Scale (HFS) developed by (Thompson et al. 2005). The self-reported 18-item instrument measures dispositional forgiveness (the general tendency to be forgiving) and has three 6-item sub-scales: forgiveness for self (e.g., "It is tough for me to accept myself once I've messed up."), forgiveness for others (e.g., "When someone disappoints me, I can eventually move past



it."), and forgiveness for situations (e.g., "I eventually make peace with bad situations in my life"). Each item is rated on a seven-point Likert scale ranging from 1 (almost always false for me) to 7 (almost always true of me). The Heartland Forgiveness Scale has demonstrated desirable psychometric properties such as convergent validity and satisfactory internal consistency. This scale has been widely administered to university students and school teachers (Kumar and Dixit 2014). The present data's reliability coefficients of forgiveness for self, others, situations, and global scale are 0.827, 0.845, 8.702, and 0.822, respectively.

*Gratitude* feeling was self-reported on a 6-item scale called The Gratitude Questionnaire (GQ-6) developed by McCullough et al. (2002). The gratitude questionnaire was designed to assess individual differences in the proneness to feel grateful toward perceived benefactors in daily life. The GQ-6 has demonstrated good psychometric properties in earlier studies conducted on academicians of the Indian population. Sumi (2017) examined the reliability and validity of the GQ-6 scale in a sample of students and found satisfactory internal consistency through Cronbach's alpha ($\alpha$ = 0.92) and test–retest reliability (r = 0.86). Each item is rated on a seven-point Likert type scale ranging from 1 (strongly disagree) to 7 (strongly agree). The reliability coefficient for the present data is 0.862.

*Spirituality* was assessed on a 16-item Daily Spiritual Experience Scale (DSES) developed by Underwood and Teresi (2002). This scale studies everyday spiritual experiences, such as admiration, serenity, giving, and receiving compassionate love daily. DSES was initially developed in health studies and has been increasingly used in other settings such as social sciences, educational institutions, and organizational settings. Currently, there are two forms of DSES: complete form 16-item scale and its shorter version 6-item scale. The 16-item scale is preferred as the shorter version has a few limitations, such as a few items being double-barreled, and the wording is not identical. DSES was found to be stable over time and internal consistency was high in African-American samples (Loustalot et al. 2006). Bailly and Roussiau (2010) also found good psychometric properties of the DSES scale. DSES has been widely used among the student population in India. The reliability coefficient of the scale for the present study is 0.948.

*Happiness* was reported on a 4-item subjective happiness scale or general happiness scale developed by Lyubomirsky and Lepper (1999). This 4-item instrument contains items that tap into subjective feelings of global happiness with one's life (e.g., "Compared to most of my peers I consider myself": 1 = less happy, 7 = more happy). The subjective happiness scale has demonstrated desirable psychometric properties such as convergent validity and satisfactory internal consistency reliability on different age groups and can be used in other cultures (Sood and Gupta 2014). The reliability coefficient of the scale for the present study is 0.851.

Second- and third-semester Grade Point Average (GPA) was used as a measure of academic performance. GPA provides ongoing academic performance, reflecting the student's ability to initiate and maintain a range of self-regulatory behaviors such as time management, emotional balance, physical fitness (Rode et al. 2005). The GPA score of each student was obtained from the university. A person designated as an examination controller was briefed about the study and assured the anonymity of respondents for the retrieval of GPA scores from university records.

## 4. Data Analysis

The proposed hypothesized model depicted in Figure 1 was examined using Analysis of a Moment Structures (AMOS) 20 software. The first step was establishing the reliability and validity of all the constructs under study. Even though all the scales were reliable and valid in previous research, the constructs were relatively new among Indian management students. Exploratory- and confirmatory-factor analyses were conducted to test the factor structure of all the seven constructs.

The data was divided into a training set and a testing set for exploratory- and confirmatory-factor analysis, respectively. The exploratory-factor analysis was conducted

using principal-component analysis with varimax rotation to extract factor structure. Lower loading (<0.40) and cross loading were excluded from the matrix.

After exploratory-factor analysis, confirmatory-factor analysis was conducted to test the measurement model using AMOS 20 software. Confirmatory-factor analysis confirmed the measurement items were loaded per the pattern revealed in the exploratory-factor analysis. The satisfactory model is established per the standards suggested by Anderson and Gerbing (1988), and the fit indices were assumed acceptable and reasonably fit by following Byrne's (2013) cut-offs.

The reliability of all the constructs was established through Cronbach's alpha and composite reliability. For the validity of the constructs, Fornell and Larcker's (1981) procedure was used. Further, PROCESS macro, as proposed by Hayes (2013), was used to test moderation and moderated-mediation relationships.

## 5. Results

At times 1 and 2, all the variables have shown satisfactory reliability (Cronbach's alpha values > 0.70), except for 'forgiveness of self' (Cronbach's alpha was 0.63), which can be assumed as moderate reliability. Confirmatory-factor analysis was conducted with the help of AMOS 20 software to validate the measurement model. The model-fit indices were reasonably good (CMIN/DF = 1.804, TLI = 0.912, CFI = 0.940 and RMSEA = 0.050), showing superior model fit. The convergent and discriminant validity were calculated using the STATS Tools package. At time 1, the composite reliability of all the constructs was satisfactory.

At time 2, the composite reliability of all the constructs, except forgiveness for situation and forgiveness for others, was more than 0.70, showing satisfactory reliability (refer to Tables 1 and 2). The Average Variance Extracted (AVE) and Maximum Shared Variance (MSV) were computed to establish convergent and discriminant validity. Again, except for forgiveness for the situation and forgiveness for others, all other constructs have established their validity. Only the AVE statistics of forgiveness for situations and forgiveness for others are 0.361 and 0.452, respectively, which are less than 0.50, but the other constructs have acceptable AVE values. The MSV values of all the constructs are less than the AVE values, showing an acceptable discriminant validity in the measurement model. The two constructs, namely, forgiveness for situation and forgiveness for others, have issues with reliability and convergent validity at time 2. The probable reasons are lack of variance among multi-dimension of forgiveness; it would be better to treat forgiveness as unidimensional for further investigation or use a smaller sample size at time 2.

As the proposed model has moderated-mediating relationships, we used the regression methods described by Hayes (2013) through PROCESS. This technique allows for a more accurate analysis of the indirect effects associated with the mediating variables, compared with Baron and Kenny's (1986) techniques, allowing for indirect effects without requiring direct impact and use of non-normally distributed data, as bootstrap re-sampling is used. Four conceptual models were developed and tested to examine the direct and indirect effects of forgiveness, gratitude, spirituality, and happiness on academic performance.

The preliminary analysis of time 1 and time 2 shows significant positive correlations among variables (see Tables 1 and 3). The magnitude of the relationships seems to be more assertive at time 2. At time 1, spirituality does not significantly impact the relationship between forgiveness, gratitude, and happiness (see Table 3). Therefore, time 2 data have been used to test all the hypotheses.

**Table 1.** The inter-construct correlations, reliabilities, and validity statistics of time 1.

| | Cronbach's Alpha | CR | AVE | MSV | Forgiveness for Self | Happiness | Gratitude | Forgiveness for Situation | Forgiveness for Others | Spirituality |
|---|---|---|---|---|---|---|---|---|---|---|
| **Forgiveness for Self** | 0.915 | 0.898 | 0.451 | 0.573 | 0.707 | | | | | |
| **Happiness** | 0.908 | 0.913 | 0.680 | 0.352 | 0.105 ** | 0.715 | | | | |
| **Gratitude** | 0.830 | 0.829 | 0.496 | 0.078 | 0.321 *** | 0.448 *** | 0.730 | | | |
| **Forgiveness for Situation** | 0.805 | 0.725 | 0.576 | 0.573 | 0.010 | 0.248 *** | 0.031 | 0.590 | | |
| **Forgiveness for Others** | 0.863 | 0.856 | 0.598 | 0.332 | 0.056 * | 0.112 ** | 0.135 ** | 0.347 *** | 0.612 | |
| **Spirituality** | 0.856 | 0.797 | 0.573 | 0.052 | 0.028 | 0.109 ** | 0.032 | −0.002 | 0.134 ** | 0.705 |
| **Performance** | - | - | - | - | 0.268 *** | 0.117 *** | 0.127 *** | 0.225 *** | 0.201 *** | 0.108 *** |

Note: CR = Composite Reliability, AVE = Average Variance Extracted, MSV = Maximum Shared Variance, Sample Size (n) = 220. The square root of the average variance extracted (AVE) is shown on the diagonal of the matrix in bold. The inter-construct correlations are shown off of the diagonal. * Significant at $p < 0.05$, ** significant at mboxemphp < 0.01, *** significant at $p < 0.00$.

**Table 2.** The inter-construct correlations, reliabilities, and validity statistics at time 2.

| | Cronbach's Alpha | CR | AVE | MSV | MaxR(H) | Forgiveness for Self | Happiness | Gratitude | Forgiveness for Situation | Forgiveness for Others | Spirituality |
|---|---|---|---|---|---|---|---|---|---|---|---|
| Forgiveness for Self | 0.630 | 0.703 | 0.573 | 0.127 | 0.940 | **0.757** | | | | | |
| Happiness | 0.886 | 0.854 | 0.664 | 0.191 | 0.958 | 0.151 ** | **0.815** | | | | |
| Gratitude | 0.842 | 0.863 | 0.613 | 0.191 | 0.967 | 0.356 *** | 0.437 *** | **0.783** | | | |
| Forgiveness for Situation | 0.725 | 0.627 | 0.361 | 0.140 | 0.969 | 0.005 | 0.313 *** | 0.071 | **0.601** | | |
| Forgiveness for Others | 0.781 | 0.688 | 0.452 | 0.140 | 0.973 | 0.076 * | 0.132 ** | 0.153 ** | 0.374 *** | **0.672** | |
| Spirituality | 0.939 | 0.933 | 0.540 | 0.035 | 0.981 | 0.059 * | 0.129 ** | 0.138 ** | −0.009 | 0.188 ** | **0.735** |
| Performance | - | - | - | - | - | 0.341 *** | 0.217 *** | 0.227 *** | 0.325 *** | 0.229 *** | 0.198 *** |

Note: CR = Composite Reliability, AVE = Average Variance Extracted, MSV = Maximum Shared Variance, Sample Size (n) = 220. The square root of the average variance extracted (AVE) is shown on the diagonal of the matrix in bold. The inter-construct correlations are shown off of the diagonal. * Significant at $p < 0.05$, ** significant at $p < 0.01$, *** significant at $p < 0.00$.

We tested hypotheses 1a, 1b, 1c, and 2 using a method described by Hayes (2013). We examined the moderating effect of spirituality on the relationship among forgiveness, gratitude, and happiness using 10,000 bootstrap samples (see Table 4). The bootstrapping process provides upper- and lower-level confidence intervals (ULCI and LLCI). If the range of LLCI and ULCI does not include zero, then the effect is significant. The interaction effect of spirituality and forgiveness for self on happiness was b = 0.6906, se = 0.3460, t = 1.9959, *p* = 0.0468, and 95% CIs (0.0098 to 1.3714). The absence of zero from the range of LLCI and ULCI shows that the interaction effect is significant, supporting hypothesis 1a. The conditional effect of spirituality on the relationship between forgiveness for self and happiness was measured at a high, medium, and low level (+1 SD, mean, and −1 SD). At the high level of spirituality, effect = 0.2266, boot se = 0.0670, and 95% CIs (0.1110 to 0.3753), which supports the assumption that a higher level of spirituality will strengthen the positive association between forgiveness for self and happiness. However, at the low

level of spirituality, effect = 0.0899, boot se = 0.0613, and 95% Cis (−0.0180 to 0.2185), so the relationship between forgiveness for self and happiness will be weak. The moderating effect of spirituality on forgiveness for others and happiness (hypothesis 1b) was not supported (b = 0.3673, se = 0.3967, t = 0.9259, *p* = 0.3552, and 95% CIs (−0.4133 to 1.1480)). Similarly, hypothesis 1c, which states spirituality as a moderator between forgiveness for situation and happiness, was also not supported, b = 0.3899, se = 0.4178, t = 0.9332, *p* = 0.3514, and 95% CIs (−0.4322 to 1.2120).

**Table 3.** PROCESS results for moderation at time 1.

| Outcome Model: Happiness | β | SE | t Value | $R^2$ |
|---|---|---|---|---|
| Constant | 1.469 | 0.443 | 3.316 ** | 0.512 |
| Gender | 0.042 | 0.036 | 1.177 | |
| Age | 0.258 | 0.566 | 0.455 | |
| Work Ex | 0.105 | 0.077 | 0.257 | |
| Education | −0.008 | 0.010 | −0.76 | |
| Family Structure | 0.01 | 0.13 | 0.06 | |
| Forgiveness_Self | 0.131 | 0.065 | 2.01 ** | |
| Forgiveness_Other | 0.375 | 0.047 | 8.009 *** | |
| Forgiveness_Situation | 0.150 | 0.072 | 2.10 * | |
| Spirituality | 0.360 | 0.120 | 2.98 ** | |
| Gratitude | 0.21 | 0.050 | 4.21 *** | |
| FS × SP | 0.258 | 0.566 | 0.455 | 0.534 |
| FO × SP | 0.042 | 0.036 | 1.177 | |
| FS × SP | 0.007 | 0.025 | 0.298 | |
| GA × SP | 0.030 | 0.020 | 0.140 | |

Note: FS = Forgiveness_Self, FO = Forgiveness_Other, FS = Forgiveness_Situation, GA = Gratitude, and SP = Spirituality. * Significant at *p* < 0.05, ** significant at *p* < 0.01, *** significant at *p* < 0.00.

**Table 4.** PROCESS results for moderation hypotheses 1 and 2 using time 2 data.

| Outcome Model: Happiness | β | SE | t Value | $R^2$ |
|---|---|---|---|---|
| Constant | 1.91 | 0.42 | 4.60 ** | 0.545 |
| Gender | −0.161 | 0.076 | −2.117 * | |
| Age | 0.308 | 0.169 | 1.824 | |
| Work Ex | 0.063 | 0.069 | 0.909 | |
| Education | −0.007 | 0.037 | −0.194 | |
| Family Structure | 0.018 | 0.091 | 0.201 | |
| Forgiveness_Self | 0.155 | 0.057 | 2.703 ** | |
| Forgiveness_Other | 0.442 | 0.051 | 8.594 *** | |
| Forgiveness_Situation | 0.150 | 0.068 | 2.194 * | |
| Spirituality | 0.370 | 0.100 | 3.91 ** | |
| Gratitude | 0.251 | 0.049 | 5.157 *** | |
| FS × SP | 0.691 | 0.346 | 1.996 ** | 0.568 |
| FO × SP | 0.367 | 0.367 | 0.926 | |
| FS × SP | 0.389 | 0.417 | 0.933 | |
| GA × SP | 0.133 | 0.015 | 9.09 *** | |
| Conditional Direct Effects of Forgiveness for Self at Mean ± 1 SD | Effect | SE | Boot LLCI | Boot ULCI |
| Low Spirituality | 0.089 | 0.061 | −0.018 | 0.218 |
| High Spirituality | 0.226 | 0.067 | 0.111 | 0.375 |
| Conditional Direct Effects of Gratitude at Mean ± 1 SD | | | | |
| Low Spirituality | −0.754 | 0.216 | 0.096 | 0.359 |
| High Spirituality | 0.754 | 0.212 | 0.103 | 0.334 |

Note: Low Spirituality = ±1 deviation below mean. High Spirituality = ±1 deviation above mean. * Significant at *p* < 0.05, ** significant at *p* < 0.01, *** significant at *p* < 0.00.

Hypothesis 2 was supported. The interaction term consisting of spirituality and gratitude was significant, b = 0.1325, se = 0.0146, t = 9.0981, *p* = 0.000, and 95% CIs (0.1039 to 0.1612). Analysis of moderation effect indicates that the relationship between

gratitude and happiness was significant for all the three levels of spirituality, namely +1 SD (b = −0.7541, se = 0.2116, Boot SE = 0.0586, and 95% Boot CIs (0.1029 to 0.3345)), Mean (b = 0.000, se = 0.2136, Boot SE = 0.0452, and 95% Boot CIs (0.1313 to 0.3081)) and −1 SD (b = 0.7541, se = 0.2156, Boot SE = 0.0669, and 95% Boot CIs (0.0965 to 0.3593). To further explore the relationship between spirituality and happiness, we used the same conditional process modeling to examine moderated mediation, as Hayes (2013) suggested, following PROCESS Macro. Spirituality was found to be moderating the relationship between forgiveness for self and happiness; however, the moderated mediated index was not significant (Index = 0.0906, Se (Boot) = 0.0578, and 95% Boot CI (−0.0190 and 0.2056)), indicating no support for Hypothesis 3a (see Table 5). Hypothesis 3b and 3c were not supported because spirituality's moderating effect on the relationships among forgiveness for others, forgiveness for the situation, and happiness was already insignificant. Favoring our expectations, hypothesis 4 was supported. The moderated mediated index was significant (Index = 0.166, Se (Boot) = 0.074, and 95% Boot CI (0.019 and 0.312). Thus, we state that spirituality and happiness moderated and mediated the relationship between gratitude and academic performance.

**Table 5.** PROCESS results for moderated-mediation hypotheses 3 and 4, dependent variable: performance using time 2 data.

| | DV: Performance | | |
| --- | --- | --- | --- |
| Independent Variable | Coeff | SE | t | p |
| Constant | 0.838 | 0.537 | 1.56 | 0.120 |
| Forgiveness for Self | 0.166 | 0.074 | 2.231 | 0.027 |
| Forgiveness for Others | 0.109 | 0.101 | 1.075 | 0.284 |
| Forgiveness for Situation | 0.142 | 0.082 | 1.741 | 0.083 |
| Spirituality | 0.139 | 0.067 | 2.082 | 0.039 |
| Gratitude | 0.235 | 0.116 | 2.021 | 0.045 |
| Happiness | 2.127 | 0.802 | 2.265 | 0.009 |
| Model Summary | R2 = 0.328, F (7113) = 13.545, $p < 0.001$ | | | |

| Indirect Effect of Forgiveness for Self on Performance through Happiness | | | |
| --- | --- | --- | --- |
| | Effect | Boot SE | Boot LLCI | Boot ULCI |
| Low Spirituality | 0.0906 | 0.0578 | −0.0190 | 0.2056 |
| High Spirituality | 0.4867 | 0.5909 | −0.6915 | 1.660 |

| Direct Effect of Forgiveness for Self on Performance | | | |
| --- | --- | --- | --- |
| Effect | SE | t | LLCI | ULCI |
| 0.9322 | 1.20 | 0.7773 | −3.332 | 1.46 |

| Indirect Effect of Forgiveness for Other on Performance through Happiness | | | |
| --- | --- | --- | --- |
| | Effect | Boot SE | Boot LLCI | Boot ULCI |
| Low Spirituality | −0.2767 | 0.2862 | −1.256 | 0.0297 |
| High Spirituality | 0.0285 | 0.1832 | −0.2218 | 0.6003 |

| Direct Effect of Forgiveness for Self on Performance | | | |
| --- | --- | --- | --- |
| Effect | SE | t | LLCI | ULCI |
| 0.9430 | 0.8232 | 1.14 | −0.6980 | 2.5841 |

| Indirect Effect of Forgiveness for Situation on Performance through Happiness | | | |
| --- | --- | --- | --- |
| | Effect | Boot SE | Boot LLCI | Boot ULCI |
| Low Spirituality | 0.0950 | 0.2598 | −0.1621 | 1.2627 |
| High Spirituality | 0.1506 | 0.3694 | −0.2039 | 1.7981 |

**Table 5.** *Cont.*

| | DV: Performance | | | |
|---|---|---|---|---|
| | Direct Effect of Forgiveness for Self on Performance | | | |
| Effect | SE | t | LLCI | ULCI |
| 2.645 | 0.9518 | 2.7789 | −0.0576 | 0.5745 |
| | Indirect Effect of Gratitude on Performance through Happiness | | | |
| | Effect | Boot SE | Boot LLCI | Boot ULCI |
| Low Spirituality | 0.085 | 0.096 | −0.091 | 0.289 |
| High Spirituality | 0.432 [a] | 0.101 | 0.236 | 0.633 |
| | Direct Effect of Gratitude on Performance | | | |
| Effect | SE | t | LLCI | ULCI |
| 0.166 | 0.074 | 2.231 | 0.019 | 0.312 |

[a] Bootstrap confidence interval for the indirect effect does not include zero.

## 6. Discussion

Recently, many psychologists have argued that research in psychology has massively ignored the positive strengths of human beings and has mainly dealt with the negative aspects, such as reducing anxiety, stress, and other kinds of psychological issues (Seligman and Csikszentmihalyi 2014). The concerns of psychologists were addressed through a new line of research called the positive-psychological movement. Positive psychology tries to identify the strengths among individuals and their impact on the various facets of life. This study also extends the positive-psychology literature by investigating the relationships between positive strengths such as forgiveness, gratitude, spirituality, and happiness. Forgiveness and gratitude were considered as the positive-psychological responses of an individual toward harm and benefits. Earlier studies found positive relationships among forgiveness, gratitude, and various measures of psychological and physiological well-being. The present study tries to extend the literature by exploring the direct and indirect effects of forgiveness and gratitude on students' academic happiness and performance.

The results support all the direct associations between forgiveness and happiness. Similarly, gratitude and happiness were found to be positively related. Both those findings are consistent with previous studies (Hill and Allemand 2011; Peterson 2015). Peterson (2015) only found a positive relationship between forgiveness of self and SWB, but the present study found a positive association between all the components of forgiveness and happiness. Thus, it can be concluded that forgiveness (self, others, and the situation) and gratitude enhances happiness among students.

Spirituality moderates the relationship between forgiveness, gratitude, and happiness. However, spirituality has not been moderating any relationships before spirituality intervention, reflecting the importance of spirituality workshops or interventions in students' lives. The moderating effect of spirituality on the relationship between forgiveness of self and happiness is established. Spirituality enhances forgiveness and acts as a protective factor against negative emotions. At the same time, the other two hypothesized relationships between forgiveness (others and situations) and happiness did not support spirituality as a moderator. The finding is consistent with Peterson (2015); they found the relationship between forgiveness and SWB will be stronger in the presence of high religious/spiritual orientation. In contrast, they did not see the moderation effect of spirituality when forgiveness was broken into self, others, and situations. They argued that individual components of forgiveness probably do not have sufficient variance compared to total forgiveness. However, the present study found the moderation effect of spirituality on the relationship between forgiveness of self and happiness. The moderated-mediated impact of spirituality and happiness on the relationship between forgiveness and academic performance was also not supported. The possible reason could be the latter two forms of forgiveness (others and situations) are more complex. They require a better understanding of an individual's

relationship with the transgressor, the emotions involved, and the motivation of the transgressor when there is no prior association (Peterson 2015). Moreover, some other variables must be incorporated into the model to understand the above relationships better.

Limited studies have explored the moderating effect of spirituality on the relationship between gratitude and happiness. However, few studies have discussed the nature of the relationship between gratitude and spirituality. For example, in qualitative research, Chao et al. (2002) reported that appreciation is one of the means to practice spirituality through giving thanks and embracing grace. Spirituality involves exploring sacred superpowers (Hill and Pargament 2003), which reflects a strong belief that some divine power is responsible for all the events in one's life. This belief enhances affection and concern for others and strengthens the feeling of gratitude. The moderated-mediation effect of spirituality and happiness on the relationship between gratitude and academic performance was also supported. This reflects that a high level of spirituality strengthens the relationship between gratitude and happiness and, further, enhanced happiness that helps students to outperform in their work.

Happiness is found to be positively related to academic performance (Seligman et al. 2009). Quinn and Duckworth (2007) established the causal relationship between happiness and academic performance in fifth-grade students. Therefore, it can be concluded that happiness is one of the strong predictors of academic success.

Management education in India is undergoing significant transitions, and management courses are becoming costlier year after year. There are numerous challenges for the administrators to address and deliver the desired value for the management students. Many business schools in India are also miserable because of poor placements. It is undoubtedly going to take substantial time to solve all these problems. In due process, the most-affected individuals are students. Paying significant tuition fees and not getting decent placements for various reasons will push students and their parents into stressful situations. Considering the scenario, building upon a student's character strengths is of paramount importance.

If students are happy, their academic performance will be enhanced and their probability of getting placed also increases. If students get good posts, then business schools' success rates will also grow regarding admissions, profits, and the ability to pay their employees. To better equip their students to face difficult situations, university administrators should also focus on their character strengths, such as forgiveness, gratitude, happiness, and spirituality. The study findings suggest important implications for students, parents, teachers, and educational institutions. First, it is essential to educate students about the importance of positive-character strengths and their influence on various aspects of life. Second, all the stakeholders involved with students should attempt to access the gratitude, spirituality, and forgiveness nature in students, for better guidance and evaluation of their happiness and performance. Third, students should be exposed to seminars, workshops, and training programs that foster a greater sense of gratitude, spirituality, and forgiveness. Fourth, as spirituality was found as a moderator, it would be helpful to add a spiritual dimension to gratitude and forgiveness; it could be done by asking students to think about the divine source regularly for the strength to forgive and to identify the acts for which they are grateful for. One of the possible techniques for encouraging students to practice forgiveness and gratitude more often is to ask them to write down weekly (or some other feasible time interval) five things for which they feel grateful and five situations where they forgive the self, other, or situation. Finally, teachers should try to incorporate relevant character strengths in their teaching, to help students in attaining the learning outcomes of the particular course as well as overall program.

## 7. Limitations

There are quite a few limitations in the present study, which future researchers can overcome. First, the sample only consists of students with very similar demographic profiles such as age and educational qualifications, which reduces the generalizability of

the study. For example, a few researchers, including Cheng and Yim (2008), suggested a differential impact of age on practicing forgiveness. They found that middle-aged adults were higher scorers on forgiveness measures. However, the present study focused only on students, which did not provide the opportunity to tap forgiveness variability per age., Second, a few studies, such as Rijavec et al. (2010), found gender-related differences in forgiveness styles. Men prefer to adopt avoidance or revenge, whereas women seek more revenge than avoidance toward transgressors. The present study did not investigate the gender-related differences because of the small sample size. Third, the present study was conducted on a cross-sectional-research design, but longitudinal research may help establish causal relationships among variables.

## 8. Conclusions

Students' happiness and well-being have received more attention, as they have been proposed to improve learning, experiences, and performance. In the present study, we contend that character strengths such as forgiveness, spirituality, and gratitude are positively related to students' happiness and academic performance. We have established that the above-mentioned character strengths affect the happiness and performance of management students in the Indian context. Precisely, it was found that spirituality moderates the relationship between forgiveness for self and happiness. Spirituality and happiness moderated and mediated the relationship between gratitude and academic performance. The findings of the study urge the stakeholders involved in students' lives and careers to understand the importance of the spirituality, gratitude, and forgiveness virtues for students. In the present scenario of continuous change, intense competition, and uncertainty, students need to strengthen these virtues for happiness and success.

**Author Contributions:** Conceptualization: R.D., Methodology: S.S., Validation of results: N.R., Final analysis; S.S., Writing—original draft and preparation: R.D., S.S. and N.R., writing—review and drafting: D.R.G. All authors have read and agreed to the published version of the manuscript.

**Funding:** This research received no external funding.

**Institutional Review Board Statement:** The current study is exempted from ethical committee.

**Informed Consent Statement:** The information provided by you will be treated as confidential and will be used in summary solely for academic purposes without disclosing your identity. Participation in this survey is voluntary; however, we hope that you will complete this questionnaire since your views are very important. I would very much appreciate your participation in this survey.

**Data Availability Statement:** In order to maintain the confidentiality, we are not willing to publish the entire data set.

**Conflicts of Interest:** The authors declare no conflict of interest.

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
