# Peer review of "Does Spirituality Influence Happiness and Academic Performance?"

_religions, doi:10.3390/rel13070617_

Round 1

Reviewer 1 Report

The article, with a detailed theoretical background, is important in that it brings together in one model spirituality, gratitude, forgiveness, and academic performance.

I consider that what is missing from this study is a paragraph in which to briefly describe the Data analysis.

Lines 240-255:  for GQ-6 and DSES questionnaires the names of the authors who found ”good psychometric properties” of the scales can also be given.

Table 3 - at a coefficient β=0.25, the value of t of 0.45 seems too small.

 As a minor errror: Likert typedddd - Line 245

Author Response

Dear Reviewer,

We sincerely thank you for your time and effort in reviewing and preparing constructive comments.

Your feedback guided us to improvise the manuscript and make it a better version.

Kindly find the table of your comments and our responses in the attachment.

Reviewer 2 Report

Feedback on Religions 1781570, 13 June 2022

General comment – This is an interesting and informative article. It has a sound Introduction and Theoretical background, with acceptable instruments employed in the study. Reporting of Results needs some finetuning, which is detailed below. The Discussion is sound. I suggest that Limitations be written in their own section with a succinct Conclusion addressing/answering the question posed by the title of the article.

Detailed comments:

Line 6 consider replacing ‘causes’ with ‘indicators of’

Line 8 rather than ‘foreseeing’, I suggest ‘revealing’ would be more accurate

Line 11 remove ‘have’ before ‘voluntarily…’

Lines 18-19 suggest rewording to ‘to assess the students’ character strengths related to their well-being and success.’

Line 20 delete ‘a’ before ‘theoretical…’ and change ‘…pointing to the existence…’

Line 29 delete ‘have’ before ‘initiated…’

Line 36 consider rewording to ‘top priority for producing…’

Line 37 replace ‘in’ by ‘on’

Line 44 ‘once they have graduated’

Line 59 small ‘l’ for ‘lion’s’

Line 61 suggest replacing ‘amplified’ with ‘grown’

Line 78 Please provide a reference for the statement regarding association of forgiveness with spirituality

Line 80 is there evidence for ‘all’ religions?

Lines 93-95 evidence/references?

Line 96 suggest you use ‘such as’ in place of ‘like’ Aristotle…

Line 97 remove ‘the’ before ‘empirical investigations…’

Line 115 replace ‘like’ with ‘such as’ nature…

Line 124 insert ‘a’ before ‘beneficiary

Line 139 begin new paragraph ‘it is also worth…’ as a new idea is presented.

Lines 162-3 Is ‘happiness’ necessarily ‘an essential factor’ or is it only ‘desirable’? There have been grumpy high achievers.

Line 165 ‘once they have graduated’, rather than ‘are’

Line 177 ‘outperform’ sits awkwardly here, possibly reword or expand

Lines 215-6 As students were identified to access their academic results, how can you be sure this did not lead to them giving more favourable responses on the post-test, as there were significant results there, which did not show in the pre-test?

Line 219 delete ‘was’ following (55%)

Line 223 suggest you replace ‘fill’ with ‘complete the same questionnaire...’

Line 225 suggest you replace ‘data points’ with ‘responses’. Also ‘The demographic profile of respondents at time 2 was very similar to those at time 1.’

Lines 273-5 suggest ‘all the variables showed satisfactory reliability (Cronbach alpha values >.70). At time 2, except for ‘forgiveness of self’ …. (…forgiveness of ‘self’ was 0.63,…).

Line 280 new paragraph beginning ‘At time 2…’ to make it easier to separate the results meaningfully.

Line 286   0.452 not ‘0.462’? Check results in Table 2 for ‘Forgiveness of others’

Lines 387-8 It might be cynical of me, but how can you be sure that the students’ responses have not simply been influenced by the workshops to provide more acceptable responses to questionnaire items once that have ‘learned the language’?

Line 433 delete ‘to’ following ‘pay’ to read ‘ ability to pay their employees.’

Line 446 delete ‘for’ following ‘grateful (as you already have ‘for’ preceding ‘which they are’

Line 448 delete ‘list’ before ‘five things’ since you ask them to ‘write down’ on line 447

Do you have https:// addresses for references 7, 23, 36, 38?

Author Response

Dear Reviewer,

We sincerely thank you for your time and effort in reviewing our manuscript word by word and providing constructive comments.

Your feedback guided us to improvise the manuscript and make it a better version.

Kindly find the table of your comments and our responses in the attachment.

Reviewer 3 Report

As far as it goes, the author chooses and researches the topic with competence. The research deals with a very sensitive and deeply personal issue. For starters, the author uses a definition of spirituality that is used widely in other research. The definition is, “Spirituality is defined as “a subjective experience of the sacred" (Vaughan, 84 1991). That is a rather standard definition and useful enough. But, there are deeper issues. It is entirely possible to have a subjective experience of the sacred and the experience might be a delusional experience of a troubled person who might be bi-polar or schizophrenic. The experience might be a narcissistic explanation of one’s own ego projections, having very little to do with spirituality as is commonly understood. There is no way to factor this out for research of this type but I think some explanation of the possible problem should be given.

Another issue that could be mentioned is the spirituality, as commonly understood, is not a utilitarian experience. Spirituality does not exist to lose weight, have better academic performance or even become personally happy. Spirituality is what it is, with no other explanation. Some spiritual persons claim a solid relationship with the sacred, call that what we will, with little personal satisfaction on this earth. It is also true that research of this type is correlational and not causative and therefore legitimate. But, again, for the purpose of this study and it's proper parameters, this issue need not dampen the research or the results. But I also think a little explanation can be given in the article.

This research is well done with a substantial bibliography, solid research design and valid conclusions. I would recommend that the article be published with the couple comments that I suggest.

Author Response

Dear Reviewer,

We are grateful for you constructive feedback on our manuscript.

Your feedback motivated us to look deeper into the theoretical understanding of Spirituality.

You comments not only guided to improvise the manuscript but help us to explore more on the spirituality construct.

Kindly find the response from our end in the attachment:
